# Development and Application of a Tandem Force Sensor

**DOI:** 10.3390/s20216042

**Published:** 2020-10-23

**Authors:** Zhijian Zhang, Youping Chen, Dailin Zhang

**Affiliations:** State Key Laboratory of Digital Manufacturing Equipment & Technology, School of Mechanical Science and Engineering, Huazhong University of Science and Technology, Wuhan 430074, China; zhijian516@hust.edu.cn (Z.Z.); ypchen@hust.edu.cn (Y.C.)

**Keywords:** tandem force sensor, traction force sensor, human–robot interaction, contact task, imitation learning

## Abstract

In robot teaching for contact tasks, it is necessary to not only accurately perceive the traction force exerted by hands, but also to perceive the contact force at the robot end. This paper develops a tandem force sensor to detect traction and contact forces. As a component of the tandem force sensor, a cylindrical traction force sensor is developed to detect the traction force applied by hands. Its structure is designed to be suitable for humans to operate, and the mechanical model of its cylinder-shaped elastic structural body has been analyzed. After calibration, the cylindrical traction force sensor is proven to be able to detect forces/moments with small errors. Then, a tandem force sensor is developed based on the developed cylindrical traction force sensor and a wrist force sensor. The robot teaching experiment of drawer switches were made and the results confirm that the developed traction force sensor is simple to operate and the tandem force sensor can achieve the perception of the traction and contact forces.

## 1. Introduction

Imitation learning or learning by demonstration is one of the promising approaches for non-experts to develop a task control method or a policy in a straightforward and feasible manner [1,2]. Within imitation learning, a task control model or policy is learned from the task demonstrations, one of which is a sequence of state-action pairs recorded during the teacher’s demonstration. After the teacher demonstrates how to complete the task several times, learning algorithms utilize the state-action pairs in these demonstrations to derive a mapping model of the state and action, namely the policy. 

To obtain the state-action pairs in demonstrations, the robot needs to sense the environment information and the actions taken by the teacher simultaneously during the task demonstration. The environment information depends on the task to be learned. In non-contact tasks of industrial robots, such as spraying and welding, the state only contains the robot motion parameters, target position, and posture, etc. [3,4]. In the contact tasks of industrial robots, the contact force needs to be included [5,6,7,8,9]. The actions taken by a teacher can be perceived by sensors, such as visual sensors to capture a teacher’s body movements [10,11] or recognize a teacher’s gestures [12], wearable sensors, and force sensors to perceive a teacher’s behavioral intentions [13,14,15]. Compared with visual sensors, wearable sensors, etc., force sensor-based kinesthetic teaching is suitable for non-professionals to tell the robot the action needed to be taken in current state in a simple and intuitive way [5,6,7,13,14,15,16].

In the robot teaching for contact tasks, force sensors need to detect not only traction force, but also the contact force. However, there is only one perceptual unit in a wrist force sensor, which makes it impossible to detect the traction and contact forces synchronously. In the imitation learning of peg-in-hole tasks, references [17,18,19] adopted kinesthetic teaching to guide the robot to carry out assembly tasks, in which a wrist force sensors was used to measure the traction force exerted by human hands and the contact status between peg and holes. However, this force sensor installed at the end flange of robot cannot distinguish between contact force and traction force, which makes the force data used for the policy learning inaccurate. To avoid this problem, Abu-Dakka [18] repeated the demonstration trajectory to collect the net contact force, which is complicated. Different from reference [18], in reference [13], Zeng grasped the end-point of Baxter robot to guide the robot motion, and the force sensor installed at the end flange of robot just detected the contact status. However, this method is only suitable for collaborative robots equipped with joint torque sensors rather than common ones. One method to obtain traction and contact forces is to adopt two wrist force sensors mounted in parallel, which can complicate the robot’s end structure [20,21]. For example, the last two joints of the robot in reference [20] cannot move freely within their motion range, which limits the adjustable range of the robot’s attitude. Therefore, for the kinesthetic teaching of robot contact tasks, simultaneous detection of traction and contact forces is still an important issue to be solved.

The main contribution of the paper is that a tandem force sensor is developed, which helps robots to learn human skills of opening and closing a drawer. A cylindrical traction force sensor that can be connected with a contact force sensor in series is developed, which is different from the common wrist force sensors [22,23,24,25,26]. Compared with these common wrist force sensors, the main novelty of the cylindrical traction force sensor is that the sensor’s side surface is sensitive to external forces rather than the lateral end surface. Besides, in the cylindrical traction force sensor, there is a central column coaxial to and within the elastic structural body (ESB), which allows other device to be connected with this sensor without influencing the measurement of the traction force. Compared with the force sensor in reference [27], the developed traction force sensor is easier to be operated by human hands and suitable for drawer switch teaching. 

## 2. Introduction to the Tandem Force Sensor

### 2.1. The Ideal Tandem Force Sensor

To realize the perception of traction and contact forces, a tandem force sensor consisting of two perceptual units connected in series is designed, as shown in Figure 1a. Figure 1a shows an ideal tandem force sensor, which helps to understand the basic perception principle of the tandem force sensor. In the ideal tandem force sensor, one perceptual unit is connected with its side surface, and the other is connected with its end surface. In the kinesthetic teaching of the robot contact tasks, the end effector is connected to the end surface of the tandem force sensor, and the human hand guides the robot’s motion by grasping the side surface of the tandem force sensor. The traction force applied to the side surface is detected by the perception unit (i.e., traction force sensor) connected with it, and another perception unit (i.e., contact force sensor) connected with the end surface is used to measure the contact force between the end-effector and external environment. Therefore, the side surface and end surface are sensitive to the traction and contact forces, respectively.

Each perceptual unit in the tandem force sensor is composed of an elastic structural body, strain type sensors pasted on the ESB, etc., and the two ESBs in it are shown in Figure 1b. The two ESBs in the tandem force sensor are connected in series, and the serial connection mode can be explained by Figure 2. The free end of the ESB for detecting traction force is connected to the side surface, and the end surface is fixed to the free end of the ESB for detecting contact force. The fixed end of the former is directly connected to the connecting flange, while the fixed end of the latter is indirectly fixed to the connecting flange through the central column. In addition, all the connections are made by screw fastening. In application, the traction force applied to the side surface will transmitted to the ESB for detecting traction force and ultimately to the connecting flange, as shown in Figure 2. The contact force exerted on the end surface will flow to the ESB for detecting contact force and to the connecting flange through central column. By adopting the connection mode shows in Figure 2, the traction and contact forces can be detected by the corresponding ESBs, and do not interfere with each other. Finally, the tandem force sensor can achieve the perception of traction and contact forces in decoupled manner.

The two perceptual units shown in Figure 2 are connected in serial structure. In principle, the two perceptual units are independent of each other, which is similar to the measurement principle of two wrist force sensors in Figure 3. The two wrist force sensors shown in Figure 3 are connected in parallel structure, which is a currently adopted method to realize the measurement of traction and contact forces. Different form this method, the two perceptual units in the tandem force sensor are connected in series so we have named the sensor shown in Figure 1 as tandem force sensor. Compared with the perception method of the traction and contact forces shown in Figure 3, the tandem force sensor is compact in structure and does not require the handle to be fixed to the sensor. Therefore, the effect of the handle gravity on the measurement accuracy of the traction force is eliminated. Moreover, the tandem force sensor does not increase the transverse structural complexity of the robot end and will not limit the motion range of the last two joints of a six degree of freedom (6-DOF) industrial robot.

### 2.2. The Developed Tandem Force Sensor

In order to simplify the realization difficulty of the tandem force sensor, this paper proposes and designs a tandem force sensor, as shown in Figure 4a. Both the wrist force sensor and the contact force sensor in Figure 1a use the end surface to sense external forces, so the wrist force sensor is used as the contact force sensor. Based on this idea, the developed tandem force sensor is different from the ideal tandem force sensor in appearance. However, the perception principle of the developed tandem force sensor is the same as the ideal tandem force sensor, that is, the perception of the traction and contact forces are achieved by the perception units connected to the side surface and end surface of the develop tandem force sensor. Moreover, the series connection mode of the two perception units in the developed tandem force sensor is consistent with the ideal tandem force sensor, as shown in Figure 4b. The difference of the traction force sensor in the developed tandem force sensor from that of in the ideal tandem force sensor lies in that the its central column is longer. Unlike the ideal tandem force sensor, limited by the size of the contact force sensor, the contact force sensor is not surrounded by the side surface of traction force sensor. Similarly, the connections of different components of the developed tandem force sensor are made by screw fastening. Besides, to achieve the series connection of the contact force sensor and traction force sensor, an intermediate connection flange is added.

To realize the tandem force sensor, we design and develop a cylindrical traction force sensor firstly. Compared with common wrist force sensors, the unique features of the cylindrical traction force sensor are that it senses external force applied to the side surface and its internal space provides adequate space for the central column. The basic structure of the ESB of the cylindrical traction force sensor is a thin-walled cylinder. The free end of the thin-walled cylinder-shaped ESB is connected with the side surface of the traction force sensor, and its fixed end is fixed to the connecting flange, as shown in Figure 2 and Figure 4b. The internal space of the ESB is not valuable for the detection of traction force. However, it is significant for the realization of the tandem force sensor. In the tandem force sensor, the central column is not only used to connect the contact force sensor, but also provides rigid support for the contact force sensor and the end effector mounted on it. Hence, the diameter of the central column should not be small, which is 32 mm in this paper. By selecting reasonable structural parameters of the cylinder-shaped ESB, enough space can be provided for the central column, which is one of the main advantages of the cylinder-shaped ESB. In addition, the internal space is also important for the arrangement of the contact force sensor and for the signal lines of the contact force sensor.

## 3. Development of the Cylindrical Traction Force Sensor

### 3.1. Architecture of the Cylindrical Traction Force Sensor

Referring the force sensor in literature [28], the cylindrical traction force sensor is designed, as shown in Figure 5a. The cylindrical traction force sensor is consisting of a cylinder-shaped elastic structural body, a connecting fitting and a shell. The cylinder-shaped ESB shown in Figure 5b is the core of the traction force sensor, and it has layer A (black area), layer B (red area) and layer C (blue area). Compared with the diaphragm type ESB [29], cross beam type ESB [30,31,32], parallel type ESB [22,33], etc., the cylinder-shaped ESB is hollow, and the free space inside can be used as the connection channel between the contact force sensor and the traction force sensor. The layer A consists of *A1*, *A2*, *A3*, and *A4*, and layer C is composed by *C1*, *C2*, *C3*, and *C4* (Figure 6)*. A1*, *A2*, *A3*, and *A4* are uniformly distributed along the circumference, the *C1*, *C2*, *C3*, and *C4* are also uniformly distributed along the circumference. In addition, the angle between *A1* and *C1* is 45 degrees, and the angle between the slots in layer A and the slots in layer C are 45 degrees or times of 45 degrees. The fixed end of cylinder-shaped ESB is fixed to the connecting fitting shown in Figure 5c by screw fastening, and the contact force sensor is fixed to the central column of it by screw fastening. Then, the connecting fitting can be fixed to the end flange of a robot and provide rigid support for the ESB and the contact force sensor. The shell shown in Figure 5d is secured to the free end of the ESB by screw fastening, and it can transfer the traction force exerted by human hands to the free end of ESB, as shown in Figure 2. 

### 3.2. Basic Force Measurement Principle of the Cylindrical Traction Force Sensor

The basic structure of the cylinder-shaped ESB can be illustrated by Figure 6. Under the traction force, the ESB will produce bending deformation and shear deformation, which will lead to the occurrence of normal stress and shear stress in the ESB. The normal stress mainly exists in layer A and layer C, which is relatively small. Therefore, the traction force sensor uses shear stress to measure traction force.

The layer A of ESB, which is used to measure the force *F_X_* along the X-axis and the force *F_Y_* along the Y-axis, consists of *A1*, *A2*, *A3*, and *A4*. When the force *F_X_* is applied on the ESB, the *A2* and *A4* will produce shear stress. The strain values of the two points on the same diameter in the outside surface of *A2* and *A4* have the same signs, as shown in Figure 7a. Besides, under the moment *M_Z_*, the *A1*, *A2*, *A3*, and *A4* will produce shear deformation. The strain values of the two points on the same diameter in the outside surface of *A2* and *A4*, respectively, have apposite signs, and the strain values of the two points on the same diameter in the outside surface of *A1* and *A3* respectively have apposite signs, as shown in Figure 7b. When the moment *M_Z_* act on the cylinder-shaped ESB, the sum of strain values of the two points on the same diameter in the outside surface of *A2* and *A4* respectively is zero. Then, by measuring the sum of strain values of the points in the outside surface of *A2* and *A4*, respectively and using this characteristic, the force *F_X_* can obtain. Similar to *F_X_*, the force *F_Y_* can be measured by measuring the sum of strain values of the points in the outside surface of *A1* and *A3*, respectively.

The layer C of ESB used for the measurement of moment *M_Z_* is composed by *C1*, *C2*, *C3*, and *C4*. Under the moment *M_Z_*, the strain values of the two points on the same diameter in the outside surface of *C1* and *C3*, respectively, own different signs, and the sign of strain values of the points in the outside surface of *C2* is opposite to that of the points on the same diameter in *C4*, as shown in Figure 8c. When the force *F_X_* or the force *F_Y_* or the combination of both is acting on the ESB, the sign of strain values of the points located at the outside surface of *C1* is the same as that of the point on the same diameter in *C3*, the same as the *C2* and *C4* (Figure 8a,b). Then, under the force *F_X_* or the force *F_Y_* or the combination of both, the difference of strain values of the two points on the same diameter in the outside surface of *C1* and *C3* or *C2* and *C4*, respectively, is zero. However, when the *F_X_*, *F_Y,_* and *M_Z_* act on the cylinder-shaped ESB, the difference of strain values of the two points on the same diameter in the outside surface of *C1* and *C3*, respectively, is not zero, same thing with *C2* and *C4*. By using this property, the moment *M_Z_* can be detected by measuring the difference of strain values of the points in the outside surface of *C1*, and *C3* and the difference of strain values of the points in the outside surface of *C2* and *C4*.

The layer B of the ESB is a ring connected to layer A and layer C respectively, and it can measure the force *F_Z_*, the moment *M_X_* and the moment *M_Y_*. Under the force *F_Z_*, layer A bears axial pressure (Figure 9a). When this axial pressure is transmitted to the layer B, there is the axial shear stress in layer B, as shown in Figure 10. Figure 10 illustrates the basic constitutional unit of the ESB, and the expansion diagram of ESB is shown in Figure 11. The axial pressure induced by force *F_Z_* causes shear deformation of *B1*, *B2*, *B3*, *B4*, *B5*, *B6*, *B7*, and *B8* (*B1−B8*), and then generate axial shear stress in the axial cross section of *B1−B8*. Moreover, the sign of strain values of the points in the outside surface of *B1−B8* are the same. Besides, under the moment *M_X_*, *A2* and *A4* are subjected to the axial pressure in opposite direction, respectively (Figure 9b), which causes shear deformation in *B3*, *B4*, *B7*, and *B8*. The sign of strain values of the points in the outside surface of *B3* and *B4*, respectively, is opposite to that of in *B7* and *B8*. Then, by using this property, the force *F_Z_* can be measured by measuring the sum of strain values of the points in the outside surface of *B1−B8*, and the moment *M_X_* can be measured by the difference between the strain values of the points in the outside surface of *B3* and *B7* and that of in *B4* and *B8*. Similar to moment *M_X_*, moment *M_Y_* can also be measured.

### 3.3. Mechanical Model of the Cylindrical Traction Force Sensor

To meet the design requirement of traction force sensor, the selection of ESB structural sizes should be carried out on the basis of theoretical analysis. Therefore, based on theory of mechanics, we analyze the mechanical properties of the ESB and establish the mechanical model of the ESB, which is of great significance for the determination of structural sizes of ESB and for the understanding of the mechanism of force perception and the mechanical properties of the ESB.

#### 3.3.1. The Mechanics Analysis under the F_X_

When the traction force *F_X_* exerts on ESB, the circular ring between layer A and the free end of the ESB will produce shear deformation along the force direction. According to the mechanics of materials, the direction of the shear stress of a point on the excircle of the circular ring coincides with the tangential direction of the excircle of it, and the angle between its direction vector and the direction of force *F_X_* is an acute angle, as shown in Figure 12. According to the calculation method of shear stress, the shear stress of point *e* can be calculated by using the following equation.
(1)$$\tau_{F_{X}}=\frac{F_{X}\cdot S_{z}}{I_{z}\cdot (D-d)/2}=\frac{4F_{X}\cdot \sin\alpha }{\pi D(D-d)}$$
where $S_{z}=D^{2}(D-d)\sin\alpha /8$ is the static moment of $\hat{ce}$ segment ring with regard to Z-axis, $I_{z}=\pi D^{3}(D-d)/16$ is the moment of inertia with respect to Z-axis, *D* is the diameter of the excircle of the ESB, *d* is the diameter of the inner circle of the ESB, α is the acute angle between point *a* and point *e* about the Z-axis.

According to Equation (1), the shear stresses of point *a* and point *c* are zero, and the shear stresses of point *b* and point *f* are the largest. Then, the distribution of shear stress values of points in the outer surface of the circular ring is shown in Figure 13.

Based on Equation (1) and Figure 13, without considering stress concentration, the distribution of shear stress values of the points in the outside surface of layer A is shown in Figure 14a. According to Figure 14a, the shear stress in *A1* and *A3* is small, while that of *A2* and *A4* is large. Therefore, the shear stress of the points in *A2* and *A4* can be utilized to measure *F_X_*. In addition, the largest shear stress value in *A2* and *A4* caused by force *F_X_* is as follows.
(2)$$\tau_{F_{X}}=\frac{4F_{X}\cdot \sin(\pi /2)}{\pi D(D-d)}=\frac{4F_{X}}{\pi D(D-d)}$$

Similar to layer A, the distribution of shear stress values of the points in the outside surface of layer C can be obtained, as shown in Figure 14b. According to Figure 14b, the values of shear stress of the points in *C1*, *C2*, *C3*, and *C4* are not too large nor too small. Besides, the direction of shear stress at points in *C2* is the same as that of the points in *C4* and the direction of shear stress at points in *C1* is the same as that of the points in *C3* (Figure 8a).

Unlike layer A and layer C, under the *F_X_*, the layer B bears no shear stress. For layer B, the shear stress in *A2* and *A4* transmits to *B34* and *B78* that connects with layer A, which induces the occurrence of the normal stress in layer B, as shown in Figure 15. This paper utilizes the shear stress in the ESB to measure the traction force. Therefore, the normal stress in layer B will not affect the measurement of *M_X_*, *M_Y_* and *F_Z_*.

#### 3.3.2. The Mechanics Analysis under the F_Y_

According to the basic structure of the ESB, under the *F_Y_*, the deformation of the ESB is similar to that of under the *F_X_*. Similar to Equation (2), the following formula is important for the measurement of force *F_Y_*.
(3)$$\tau_{F_{Y}}=\frac{4F_{Y}}{\pi D(D-d)}$$

However, unlike under force *F_X_*, the points with the largest shear stress are in *A1* and *A3* and the points that owns zero shear stress value exist in *A2* and *A4*. Hence, the indirect measurement of force *F_Y_* can be achieved by using the shear stress values of the points in *A1* and *A3*.

#### 3.3.3. The Mechanics Analysis under the Force F_Z_

Under the force *F_Z_*, the *A1*, *A2*, *A3*, and *A4* bear axial pressure, the *C1*, *C2*, *C3*, and *C4* also under axial pressure. Therefore, the shear stress of the points in the outside surface of layer A and layer C is zero. According to Figure 10 and Figure 11, under the *F_Z_*, the cross sections along Z-axis of *B1*−*B8* will bear shear force, and the shear stress of the points in *B1−B8* can be calculated using the following equation.
(4)$$\tau_{F_{Z}}=\frac{F_{Z}}{A}=\frac{2F_{Z}}{L_{b}(D-d)}$$
where $A=L_{b}(D-d)/2$ is the area of the cross section along Z-axis of layer B (Figure 10), (D−d)/2 is the wall thickness of layer B, $L_{b}$ is the height of layer B.

Then, based on Equation (4), the force *F_Z_* can be measured by detecting the shear stress values of the points in *B1−B8*.

#### 3.3.4. The Mechanics Analysis under the Moment M_X_

When the moment *M_X_* acts on the cylinder-shaped ESB, the force/moment applied on *A1*, *A2*, *A3*, and *A4* can be simplified as shown in Figure 9b, which leads to the occurrence of normal stress in the *A1*, *A2*, *A3*, and *A4*. In addition, under the *M_X_*, *C1*, *C2*, *C3*, and *C4* will also produce normal stress, but no shear stress. When the *M_X_* is positive, *A2* bears the largest tension and *A4* understands the largest pressure. However, the normal stress in *A1* and *A3* is close to zero, because the central axis of twist goes through *A1* and *A3*. For the layer B, the tension applied on *A2* will transmit to *B34*, and the tension in *B34* will cause shear stress in the outside surface of *B3* and *B4*. Similarly, the outside surface of *B7* and *B8* will also produce shear stress. Because the normal stress in *A1* and *A3* is approximately zero, the tensions/pressures applied on *B3* and *B4* or *B7* and *B8* induced by moment *M_X_* approximate to $F_{M_{X}}=M_{X}/D$. Then, the largest shear stress value of the points in the outside face of *B3*, *B4*, *B7* and *B8* can be figured out.
(5)$$\tau_{M_{X}}=\frac{F_{M_{X}}}{A}=\frac{M_{X}/D}{L_{b}(D-d)}=\frac{M_{X}}{L_{b}(D-d)D}$$

Although both *F_Z_* and *M_X_* cause shear stress in *B3*, *B4*, *B7*, and *B8*, the sign of shear stress incurred by *M_X_* in *B3* and *B4* is apposite to that of *B7* and *B8*, the sign of shear stress caused by *F_Z_* in *B3* and *B4* is the same as that of *B7* and *B8*. Then, the shear stress of the points in the outside surface of *B3* and *B4* minus the shear stress of the points in the outside surface of *B7* and *B8* is the shear stress caused by *M_X_*. On the contrary, the shear stress of the points in the outside surface of *B3* and *B4* add the shear stress of the points in the outside surface of *B7* and *B8* is the shear stress caused by *F_Z_*. Therefore, by using this property, *F_Z_* and *M_X_* can be measured, respectively.

In addition, the *F_Y_* applied on the ESB generates the moment around X-axis at layer B, as shown in Figure 16. Therefore, the moment measured by using the shear stress in *B3*, *B4*, *B7*, and *B8* is the superposition of the true moment *M_X_* and the moment caused by *F_Y_*. However, the true moment *M_X_* applied on the ESB is the moment value we need to measure. The force *F_Y_* is measurable by using the shear stress in *A1* and *A3*, and the moment arm of the moment caused by *F_Y_* is available. Then, the moment caused by *F_Y_* can be calculated out, after which the true moment *M_X_* is obtainable.

#### 3.3.5. The Mechanics Analysis under the M_Y_

Under the *M_Y_*, the deformation of ESB is similar to that of under the *M_X_*. Therefore, similar to Equation (5), the following equation can be obtained.
(6)$$\tau_{M_{Y}}=\frac{M_{Y}}{L_{b}(D-d)D}$$

Unlike under moment *M_X_*, under the *M_Y_*, *A1*, and *A3* bear the largest tension or pressure, and the normal stress in *A2* and *A4* is close to zero. Then, the points in the outside surface of *B1*, *B2*, *B5*, and *B6* produce relatively large shear stress. Under the combined action of *M_Y_* and *F_Z_*, both will cause shear stress in *B1*, *B2*, *B5*, and *B6*. The sign of shear stress incurred by *M_Y_* in *B1* and *B2* is apposite to that of *B5* and *B6*, the sign of shear stress caused by *F_Z_* in *B1* and *B2* is the same as that of *B5* and *B6*. By using this property, the *F_Z_* and *M_X_* can be measured respectively. In addition, the *F_X_* applied on the shell will also produce moment around Y-axis. Therefore, the measurement of true moment *M_Y_* applied on the ESB also needs to wipe off the moment around Y-axis caused by *F_X_*.

#### 3.3.6. The Mechanics Analysis under the Moment M_Z_

When the moment *M_Z_* act on the cylinder-shaped ESB, the *A1*, *A2*, *A3*, and *A4* all produce shear stress and the value of shear stress can be calculated using the following equation.
(7)$$\tau_{M_{Z}}=\frac{M_{Z}}{R\cdot A^{\prime }}=\frac{M_{Z}}{D/2\cdot \pi D(D-d)/2(r+1)}=\frac{4M_{Z}(r+1)}{\pi D^{2}(D-d)}$$
where $A^{\prime }=\pi D(D-d)/2(r+1)$ is the area of the cross section of layer A perpendicular to Z-axis, R=D/2 is the radius of the excircle of the ESB, *r* is the ratio between the arc length of four grooves and the arc length of *A1*, *A2*, *A3* and *A4*.

Under the *M_Z_*, the direction of shear stress of the points in the outer surface of *A1* and *A2* is opposite to that of *A3* and *A4* respectively, as shown in Figure 7b. However, under the *F_X_*, the direction of shear stress of the points in *A2* and *A4* is the same (Figure 7a). Similarly, under the *F_Y_*, the direction of shear stress of the points in *A1* and *A3* is the same. Therefore, the measurement of *F_X_* or *F_Y_* that using the shear stress of the outside surface of *A2* and *A4* or *A1* and *A3* respectively will not be affected by *M_Z_*.

For layer B, the shear stress in *A1*, *A2*, *A3*, and *A4* transmits to *B12*, *B34*, *B56*, and *B78* that connects with layer A. Then, *B1*−*B8* under the normal stress, which does not affect the measurement of *M_X_*, *M_Y_* and *F_Z_*. The normal stress in *B1*−*B8* causes stress in *B23*, *B45*, *B67*, and *B81*, which induces the shear stress in *C1*, *C2*, *C3*, and *C4*. The values of the shear stress in the outside surface of *C1*, *C2*, *C3*, and *C4* are the same as that of layer A, which can be figured out by using Equation (7). Similarly, the direction of the shear stress of the points in the outside surface of *C1* is opposite to that of *C3*, the direction of the shear stress of the points in the outside surface of *C2* is opposite to that of *C4* (Figure 8c). Besides, *F_X_* and *F_Y_* also affect the shear stress of the points in the outside surface of *C1*, *C2*, *C3*, and *C4*. However, according to Figure 8a,b, under *F_X_* and *F_Y_*, the direction of the shear stress of the points in the outside surface of *C1* is the same as that of *C3* and the shear stress of the points in the outside surface of *C2* is the same as that of *C4*. Hence, the shear stress in the outside surface of *C1*, *C2*, *C3*, and *C4* can be used to detect *M_Z_* without being affected by *F_X_* and *F_Y_*.

### 3.4. Parameter Selection and Strength Check of Cylinder-Shaped Elastic Structural Body

#### 3.4.1. Sensitivity and Parameter Selection of the Elastic Structural Body

Strain values under unit forces and torques can reflect the sensitivities of a force sensor. The microstrain measured by the strain gauge is ε=τ/E, where E is the elasticity modulus and τ is the shear stress caused by unit force/moment. Aluminum alloy 7075 is selected to machine the cylinder-shaped ESB, and the elasticity modulus of aluminum alloy 7075 is E=71.7 Gpa. Under the unit traction force, the micro strains measured by the strain gauges pasted in the outside surface of the ESB are as follows.
(8)$$\left\{ \begin{array}{l} S_{F_{X}}=\;S_{F_{Y}}=4/\pi D(D-d)E\\ S_{F_{Z}}=2/l_{b}(D-d)E\\ S_{M_{X}}=\;S_{M_{Y}}=1/l_{b}(D-d)DE\\ S_{M_{Z}}=4(r+1)/\pi D^{2}(D-d)E \end{array}\right.$$where $S_{F_{X}},S_{F_{Y}},S_{F_{Z}},S_{M_{X}},S_{M_{Y}},S_{M_{Z}}$ are the sensitivities of ESB with respect to traction forces/torques $F_{X},F_{Y},F_{Z},M_{X},M_{Y},M_{Z}$ respectively.

According to Equation (8), the smaller *D*, *d* and $l_{b}$ are, the larger r is, and the larger sensitivities will be. However, based on the design criteria, the parameters of the ESB must not be too small. Considering the convenience of mechanical processing of the ESB and the pasting of strain gauges, the selected parameters are D=50 mm, d=48 mm, $l_{b}=9\;mm$, r=3 and the heights of layer A and layer C are $l_{a}=10\;mm$, $l_{c}=9\;mm$, respectively. Substituting the parameters into Equation (8), the theoretical sensitivities of the ESB can be obtained, as shown in Table 1.

#### 3.4.2. Strength Check of the Cylinder-Shaped ESB

In order to prevent the overload damage of the traction force sensor, it is necessary to obtain the maximum force that the cylinder-shaped ESB can withstand, which can be calculated by the following equation.
(9)$$\left\{ \begin{array}{l} F_{X-\max}=\;F_{Y-\max}=\left(\pi D(D-d\right)/4)[\tau ]\\ F_{Z-\max}=\left(L_{b}(D-d\right)/2)\left[\tau \right]\\ M_{X-\max}=\;M_{Y-\max}=\;L_{b}(D-d)D\left[\tau \right]\\ M_{Z-\max}=\;\left(\pi D^{2}(D-d\right)/4(r+1))\left[\tau \right] \end{array}\right.$$
where [τ] is the permissible shear stress of the 7075 aluminum alloy used to machine the cylinder-shape ESB, [τ]=0.5[σ], $\left[\sigma \right]=\sigma_{s}/2.5=182\;N/mm^{2}$, $\sigma_{s}=455\;N/mm^{2}$ is the yield stress of 7075 aluminum alloy, $F_{X-max}$, $F_{Y-max}$, $F_{Z-max}$, $M_{X-max}$, $M_{Y-max}$, and $M_{Z-max}$ are the largest *F_X_*, *F_Y_*, *F_Z_*, *M_X_*, *M_Y_*, and *M_Z_* that the ESB can bear, respectively.

Substituting the parameters into Equation (9), the maximum forces/moments that the ESB can withstand can be figured out, and they are $F_{X-max}=2857.4\;N$, $F_{Y-max}=2857.4\;N$, $F_{Z-max}=819\;N$, $M_{X-max}=8190\;N\cdot cm$, $M_{Y-max}=8190\;N\cdot cm$ and $M_{Z-max}=8929.38\;N\cdot cm$, respectively. In the kinesthetic teaching of robot, humans will not to use large forces to guide the movement of robots. Therefore, for human, the theoretical maximum forces/moments that the ESB can bear are very large, which are enough to prevent the traction force sensor from damaging.

### 3.5. Measurement of the Traction Force

Given the true sensitivities of the traction force sensor, the traction force can be calculated according to the measured strain values. By combining Equations (2)−(8), the traction force can be figured out, as follows.
(10)$$\left[\begin{array}{l} F_{X}\\ F_{Y}\\ F_{Z}\\ {M^{\prime }}_{X}\\ {M^{\prime }}_{Y}\\ M_{Z} \end{array}\right]=\left[\begin{array}{cccccc} 1/S_{F_{X}} & 0 & 0 & 0 & 0 & 0\\ 0 & 1/S_{F_{Y}} & 0 & 0 & 0 & 0\\ 0 & 0 & 1/S_{F_{Z}} & 0 & 0 & 0\\ 0 & 0 & 0 & 1/S_{M_{X}} & 0 & 0\\ 0 & 0 & 0 & 0 & 1/S_{M_{Y}} & 0\\ 0 & 0 & 0 & 0 & 0 & 1/S_{M_{Z}} \end{array}\right]\left[\begin{array}{l} \varepsilon_{F_{X}}\\ \varepsilon_{F_{Y}}\\ \varepsilon_{F_{Z}}\\ \varepsilon_{{M^{\prime }}_{X}}\\ \varepsilon_{{M^{\prime }}_{Y}}\\ \varepsilon_{M_{Z}} \end{array}\right]$$
where $\varepsilon_{F_{X}}$, $\varepsilon_{F_{Y}}$, $\varepsilon_{F_{Z}}$, $\varepsilon_{M_{X}^{’}}$, $\varepsilon_{M_{Y}^{\prime }}$ and $\varepsilon_{M_{Z}}$ are the shear strain values caused by *F_X_*, *F_Y_*, *F_Z_*, ${M^{\prime }}_{X}$, ${M^{\prime }}_{Y}$, and *M_Z_*, respectively.

In order to measure the shear strains caused by external forces/moments, the strain gauges need to be pasted to the ESB in a manner of ±45° with the axis of the ESB and the strain gauges pasted in different regions are formed into six electric bridges. The output of an electric bridge is voltage, not strain value. Then, Equation (10) can be rewrote to exhibit the mapping relation between the voltage changes of electric bridges and the external forces.
(11)$${\left[F\right]}_{6\times 1}\begin{array}{cc} = & {\left[S\right]}_{6\times 6}\cdot {\left[K\right]}_{6\times 6}\cdot {\left[\Delta v\right]}_{6\times 1}\begin{array}{cc} = & {\left[P\right]}_{6\times 6}{\left[\Delta v\right]}_{6\times 1} \end{array} \end{array}$$
where ${\left[F\right]}_{6\times 1}$ is ${\left[F_{X},F_{Y},F_{Z},{M^{\prime }}_{X},{M^{\prime }}_{Y},M_{Z}\right]}^{T}$; ${\left[S\right]}_{6\times 6}$ is the diagonal matrix in Equation (10); ${\left[K\right]}_{6\times 6}$ is the coefficient matrix of strain transfer of electric bridge, the elements in ${\left[K\right]}_{6\times 6}$ is the strain values corresponding to unit voltage; the elements in ${\left[\Delta v\right]}_{6\times 1}$ are the change values in the output voltage of the electric bridges; ${\left[P\right]}_{6\times 6}$ is equal to ${\left[S\right]}_{6\times 6}{\left[K\right]}_{6\times 6}$.

Owing to the moment ${M^{\prime }}_{X}$ in ${\left[F\right]}_{6\times 1}$ includes the moment caused by *F_Y_* and the moment ${M^{\prime }}_{Y}$ contains the moment induced by *F_X_*, the amendment is necessary to get the real moment *M_X_* and *M_Y_* applied on the traction force sensor. The following equation can eliminate the errors in ${M^{\prime }}_{X}$ and ${M^{\prime }}_{Y}$.
(12)$$\left\{ \begin{array}{l} M_{X}={M^{\prime }}_{X}-F_{Y}\cdot d_{F_{Y}}\\ M_{Y}={M^{\prime }}_{Y}-F_{X}\cdot d_{F_{X}} \end{array}\right.$$
where $d_{F_{X}}$ is the moment arm from the application point of the force *F_X_* to the moment measuring point, $d_{F_{Y}}$ is the moment arm from the application point of the force *F_Y_* to the moment measuring point, and in ideal circumstances, $d_{F_{X}}$ is equal to $d_{F_{Y}}$.

### 3.6. The Realization of Traction Force Sensor

After the traction force sensor is machined, it is necessary to paste strain gauges for the shear stress measurement on the surface of the cylinder-shaped ESB. To measure the traction force, we pasted 48 miniature strain gauges on the outer surface of the cylinder-shaped ESB, and the distribution diagram of these strain gauges is shown in Figure 17. The blue rectangles in Figure 17 represent strain gauges, and the red squares in Figure 17 represent the connecting terminals of strain gauges. In order to measure the shear stress of one point, two strain gauges are pasted on the same area at an angle of 90°, and the angle between the two strain gauges and the direction of shear stress is 45° and −45° respectively. Therefore. One strain gauge is used to detect tensile stress caused by shear stress, and the other is used to measure compression stress induced by shear stress. Moreover, the strain gauges pasted in *A1*, *A2*, *A3*, *A4*, *C1*, *C2*, *C3*, and *C4* should be pasted in the area that bears the largest shear stress under the *F_X_*, *F_Y,_* and *M_Z_*, that is, the middle position of these areas. However, based on the analysis in Section 3.3.3, Section 3.3.4 and Section 3.3.5, the strain gauges pasted in *B1*−*B8* can be arranged as Figure 17 shows. After the strain gauges were pasted, the cylinder-shaped ESB is shown in Figure 19a.

After the pasting of the strain gauges, the strain gauges pasted in different areas are connected to form six electric bridges. The four strain gauges pasted in *A2* and *A4* are connected to form the 1st electric bridge to measure the strain caused by the force *F_X_*. The strain gauges pasted in *A1* and *A3* are connected to form the 2nd electric bridge to measure the strain caused by the force *F_Y_*. The strain gauges pasted in *B1_2_*, *B2_1_*, *B3_2_*, *B4_1_*, *B5_2_*, *B6_1_*, *B7_2_*, and *B8_1_* are connected to form the third electric bridge to measure the strain induced by the force *F_Z_*. Similarly, the strain caused by the moment ${M^{\prime }}_{X}$ can be measured by the fourth electric bridge made up of strain gauges stuck in *B3_1_*, *B4_2_*, *B7_1_*, and *B8_2_*, the indirect measurement of the moment ${M^{\prime }}_{Y}$ is obtainable by the fifth electric bridge made up of strain gauges pasted in *B1_1_*, *B2_2_*, *B5_1_*, and *B6_2_*, the strain caused by the moment *M_Z_* can be measured by the sixth electric bridge made up of strain gauges pasted in *C1*, *C2*, *C3*, and *C4*.

As presented in Section 3.2 and Section 3.3, this paper utilizes the sum or the difference of strain values of the points in the outside surface of the ESB to measure the traction force. The sum of strain values of the points in the outside surface of *A2* and *A4* respectively is used to represent the force *F_X_*. Therefore, the connection mode of the four strain gauges pasted in *A2* and *A4* is shown in Figure 18a. $\Delta R_{F_{X}}$ and $\Delta R_{M_{Z}}$ represent the changes in the resistance values of the strain gauges caused by the force *F_X_* and the moment *M_Z_* respectively. In addition, the minus sign and plus sign of $\Delta R_{F_{X}}$ and $\Delta R_{M_{Z}}$ indicate that the strain gauge is compressed and stretched respectively. According to the measurement principle of electric bridges, when $\Delta R_{F_{X}}$ is zero, the output voltage is zero even if $\Delta R_{M_{X}}$ is not equal to zero. However, the output voltage is not zero when $\Delta R_{M_{X}}$ is equal to zero and $\Delta R_{F_{X}}$ is not zero. Therefore, the 1st electric bridge can measure the force *F_X_*. In order to measure the forces *F_Y_* and *F_Z_*, the connection mode of the second and the third electric bridges is basically the same as that of the 1st electric bridge.

According to Section 3.2 and Section 3.3.4, the difference between the strain values of the points in the outside surface of *B3* and *B4* and that in *B7* and *B8* is used to represent the moment *M_X_*. Therefore, the connection mode of the eight strain gauges pasted in *B3_1_*, *B4_2_*, *B7_1_*, and *B8_2_* is shown in Figure 18b. $\Delta R_{F_{Z}}$ and $\Delta R_{M_{X}}$ represent the changes in the resistance values of the strain gauges caused by the force *F_Z_* and the moment *M_X_* respectively. According to the measurement principle of electric bridges, when $\Delta R_{M_{X}}$ is zero, the output voltage is zero even if $\Delta R_{F_{Z}}$ is not equal to zero. However, the output voltage is not zero when $\Delta R_{F_{Z}}$ is equal to zero and $\Delta R_{M_{X}}$ is not zero. Therefore, the fourth electric bridge can measure the moment *M_X_*. In order to measure the moments *M_Y_* and *M_Z_*, the connection mode of the fifth and the sixth electric bridges is basically the same as that of the fourth electric bridge.

According to the structure of the traction force sensor shown in Figure 5, the cylinder-shaped ESB, connecting fitting and shell were assembled into a traction force sensor by screw fastening, as shown in Figure 19b. The central column in the connecting fitting attaches the contact force sensor to the end of the traction force sensor and form a tandem force sensor. The output signal of the traction force sensor is voltage, and we developed a 12-channel signal acquisition instrument to realize the signal acquisition (Figure 19c). The 6-channel in the signal acquisition instrument is used for the signal acquisition of the traction force sensor, and the other 6-channel is used for the information acquisition of the contact force sensor.

### 3.7. Calibration Experiment of Cylindrical Traction Force Sensor

Equations (11) and (12) exhibit that the traction force can be detected by measuring variation values of voltages of six electric bridges. To obtain the real matrix ${\left[P\right]}_{6\times 6}$ in Equation (11) and the moment arm in Equation (12), calibration experiment is necessary. In the calibration experiment of traction force sensor, we use a 6-DOF industrial robot to finish the calibration experiment, as shown in Figure 20. In the calibration experiment, the robot remains stationary during the calibration process to provide a rigid support for the sensor, and forces and torques are applied to the sensor by mounting weights on the loading structure. Moreover, the attitude of the traction force sensor can be changed by adjusting the posture of the robot so that the forces/moments in different directions can be applied to the sensor. After applying forces/torques to the sensor, the self-developed signal acquisition instrument collects the output voltages of the sensor.

In the calibration experiment, small force/moment ranges are adopted because humans like to guide robot with small forces/torques. During the calibration process, *F_X_* and *F_Y_* adopt interval load of ±60 N ×10 N, *F_Z_* adopts interval load of 60 N ×10 N, *M_X_* and *M_Y_* adopt interval load of ±201 N·cm ×(10 N ×3.35 cm) and *M_Z_* adopts interval load of ±192 N·cm ×(10 N ×3.2 cm). The moments applied to the sensor were achieved by mounting weights on the loading structure. Therefore, when moments were applied to the sensor, the weights will also exert forces on the sensor. After each loading, the output values of electric bridge were recorded. Each calibration experiment was repeated three times to ensure the availability and repeatability of the experimental data. Under the external force, the changes of output voltage value are shown in Figure 21, and CH1, CH2, CH3, CH4, CH5, and CH6 represent the output voltage values of the first, second, third, fourth, fifth, and sixth electric bridge, respectively. Under the *F_X_* and *M_Z_*, CH1, CH5, and CH6 have significant output, and this certifies that *F_X_* will induce the occur of moment around Y-axis; under the force *F_Y_* and moment *M_Z_*, CH2, CH4 and CH6 have significant output, and this certifies that *F_Y_* will induce the occur of moment around X-axis. In addition, Figure 21 shows CH1 is mainly sensitive to *F_X_*, CH2 is mainly sensitive to *F_Y_*, CH3 is mainly sensitive to *F_Z_*, CH4 is mainly sensitive to *M_X_*, CH5 is mainly sensitive to *M_Y_* and CH6 is mainly sensitive to *M_Z_*. All of this certifies the theoretical analysis in Section 3.2.

After the calibration experiment, the least square method was used to calculate the calibration matrix ${\left[P\right]}_{6\times 6}$, as follows.
(13)[P]6×6=(−1.07×10^−15.21×10^−3−6.22×10^−3−2.26×10^−25.23×10^−11.37×10^−1−2.45×10^−35.44×10^−2−1.80×10^−32.20×10^−1−1.74×10^−1−4.14×10^−2−6.06×10^−43.13×10^−4−9.07×10^−2−3.50×10^−25.61×10^−21.61×10^−2−3.67×10^−35.91×10^−3−1.03×10^−2 2.621.43×10^−1−4.05×10^−2−5.64×10^−33.65×10^−32.88×10^−4−5.48×10^−22.598.91×10^−3−6.65×10^−4−7.34×10^−42.89×10^−36.99×10^−2−9.18×10^−2−2.12)

Plug the calibration matrix into Equation (11) and using Equation (12), the calculated forces/torques can be obtained, which are presented in Figure 22. Then, the interference errors of the cylindrical traction force sensor are shown in Table 2, which shows that most of the errors are not larger than 1.0%, and the measurement ranges are $-60\leq F_{X}\leq 60\;N$, $-60\leq F_{Y}\leq 60\;N$, $0\leq F_{Z}\leq 60\;N$, $-201\leq M_{X}\leq 201\;N\cdot cm$, $-201\leq M_{Y}\leq 201\;N\cdot cm$, $-192\leq M_{Z}\leq 192\;N\cdot cm$, respectively.

Non-linear errors (NLES), hysteresis errors (HES) and repeatability errors (RES) are important indexes to show the static performance of a sensor. Five of the six NLES of the cylindrical traction force sensor are not larger than 0.70%, five of the six HES are not larger than 0.85% and four of the six RES are not larger than 0.80%, as Table 3 shows. To demonstrate the measurement error visually of the sensor, several load and measurement experiments of forces/moments were conducted, and Table 4 compares the calculated values with the actual values. The measurement errors in Table 4 verified that the cylindrical traction force sensor can detect the external forces/torques applied to it, and the measurement errors are small.

## 4. The Realization and Application of the Tandem Force Sensor

### 4.1. The Tandem Force Sensor Based on the Developed Cylindrical Traction Force Sensor

According to the schematic diagram of the structure of the tandem force sensor shown in Figure 4a and the series connection mode shown in Figure 4b, a tandem force sensor is developed, as shown in Figure 23. The tandem force sensor is composed of a developed cylindrical traction force sensor and a contact force sensor connected in series, and the contact force sensor is connected with the cylindrical traction force sensor by an intermediate connecting flange. In addition, all connections are made by screw fastening. In the application, the tandem force sensor is connected to the robot end through the connection flange, and the end-effector can be fixed to the end of the tandem force sensor. In the kinesthetic teaching of robot contact tasks, the human hand exerts the traction force by grasping the shell of the traction sensor to guide the robot’s motion, while the contact force sensor can accurately perceive the contact force between the robot’s end-effector and the environment. Then, the traction and contact forces can be simultaneously perceived by the developed tandem force sensor in a decoupled manner.

### 4.2. Application of the Developed Tandem Force Sensor

To further test the feasibility of the developed tandem force sensor, this paper designs drawer switch experiment based on human–robot interaction. In daily work and life, people can easily open a variety of drawers. However, it is not an easy task for robot to open and close diversified drawers like what human does. Human–robot interaction helps to transmit experience to robot and inform the robot of the method of opening and closing drawers, and then robot can learn the method to open and close drawers.

With the developed tandem force sensor, the drawer switch experiment can complete with human–robot interaction without damaging the drawer, and the robot can obtain several effective demonstrations. In the human–robot interaction to finish the drawer switch experiment, the tandem force sensor is mounted at the end of the robot and vacuum chuck, which allows the robot to control the opening and closing of drawers, is attached to the contact force sensor. In human–robot interaction, the teacher chooses a drawer in the locker and selects the adsorption area of the drawer. People guide the robot move from the initial point to the selected drawer, and control the suction cup to hold the drawer. Then, the human guides the robot to open the drawer to the maximum and finally, the human guides the robot to close the drawer, as shown in Figure 24. In this process, the tandem force sensor detects the traction force and the contact force between vacuum chuck and drawer, which allows the robot to act according to human intentions and its contact state with the object being operated on, not just human intentions. During the experiment, the data sampled by the tandem force sensor and the action taken by the teacher are saved as state-action pairs. Then, the robot can learn the policy of drawer switch task and perform drawer opening and closing by itself (Figure 25), which confirms the feasibility and effectiveness of the tandem force sensor.

In the human–robot interaction, to complete the drawer switch experiment, the change curves of $F_{Z}^{T}$ and $F_{Z}^{C}$ (superscript *T* and *C* represent the traction force and contact force sensors, respectively) are shown in Figure 26. Owing to the inaccuracy of manual operation, the numerical fluctuation of the traction force $F_{Z}^{T}$ is high, while the numerical fluctuation of the contact force $F_{Z}^{C}$ is lower than $F_{Z}^{T}$. In order to simulate the force curve in the kinesthetic teaching of drawer switch experiment based on single wrist force sensor, the resultant force of the traction force $F_{Z}^{T}$ and the contact force $F_{Z}^{C}$ has been calculated, as shown in Figure 27. By comparing Figure 26b and Figure 27, it can be seen that the resultant force cannot accurately represent the contact state between the robot and the drawer. If the task policy is learned based on the resultant force, it cannot make the right decision. For example, when the drawer switch task policy is learned based on the data shown in Figure 27, the learned policy only outputs effective action instructions when the absolute value of contact force is about 20 *N*, instead of 0 *N*. Therefore, the net contact force obtained by the tandem force sensor is necessary for learning effective contact task policy.

## 5. Conclusions

A tandem force sensor for measuring the traction and contact forces is introduced in this paper. In cases that a wrist force sensor is used as the contact force sensor, a cylindrical traction force sensor that is easy to handle by hand, has been designed. As the core of the cylindrical traction force sensor, the cylinder-shaped elastic structural body is designed, and the force measurement theory of it is analyzed in detail. Calibration experiments verify the theoretical analysis of the cylinder-shaped elastic structural body and the good static characteristics of the traction force sensor. Then, a wrist force sensor is mounted to the developed cylindrical traction force sensor to realize the tandem force sensor.

To verify whether the tandem force sensor can meet the original intention, the drawer switch experiment based on the tandem force sensor has been carried out. The traction force sensor in drawer switch experiment transmits the human intention to the robot, and the contact force sensor detects the contact status between the robot and the drawer. Human–robot interaction experiment shows that the tandem force sensor can sense the way and skill of a teacher and the contact force between robot and environment, so that the human and robot can cooperate to complete the task, which is the basis of how robots learn to accomplish contact tasks.

The traction force sensor can be combined with the contact force sensor as a tandem force sensor, or can be used alone. Note that, although we have only applied the tandem force sensor to the drawer switch experiment, this sensor can also be applied to a wide range of contact tasks that needs human–robot collaboration, such as assembly, grinding, polishing, and deburring. Moreover, the traction force sensor can be used alone for the non-contact tasks that need human–robot collaboration, such as paint spraying and track teaching. In these tasks, the most important advantage of the traction force sensor over the common wrist force sensor is that the gravity of the end-effector does not affect its measurement, which simplifies the sensor’s gravity compensation.

Because the contact force sensor adopted in the tandem force sensor is a commercial wrist force sensor, and its structure is not optimized for the tandem force sensor, which makes the appearance of the developed tandem force sensor less graceful and complex. In the future, the structure of the tandem force sensor will be optimized, which will make it very graceful and close to the ideal tandem force sensor. By that time, the tandem force sensor can be utilized to robotic contact tasks in a very elegant way.

## Figures and Tables

**Figure 1 sensors-20-06042-f001:**
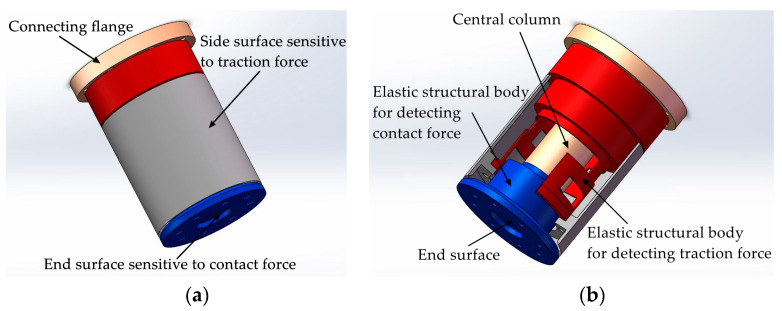
Schematic diagram of the ideal tandem force sensor: (**a**) structure of the ideal tandem force sensor; (**b**) the inner structure of the ideal tandem force sensor.

**Figure 2 sensors-20-06042-f002:**
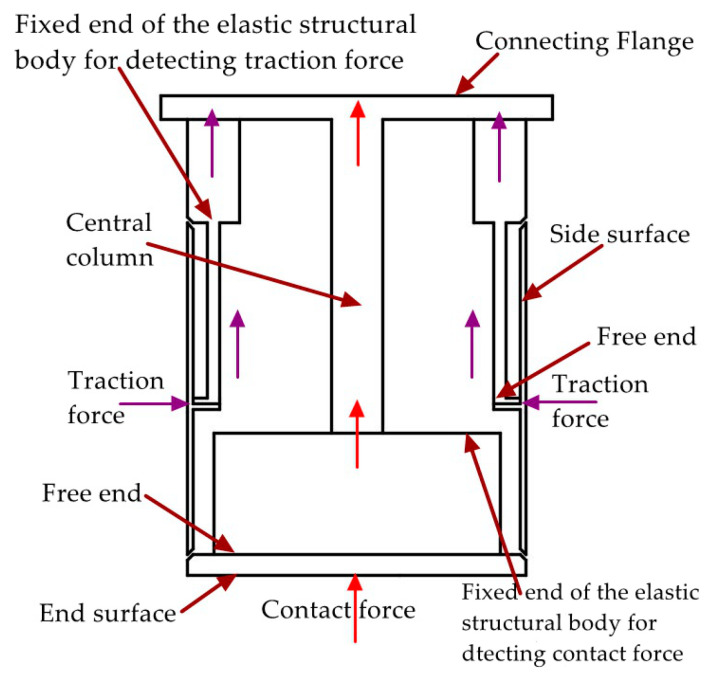
Simplified schematic diagram of series connection mode of the tandem force sensor.

**Figure 3 sensors-20-06042-f003:**
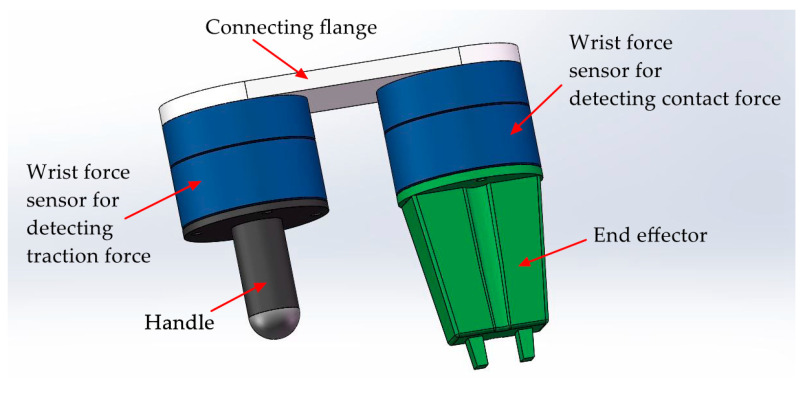
Two wrist force sensors installed in parallel for detecting the traction and contact forces.

**Figure 4 sensors-20-06042-f004:**
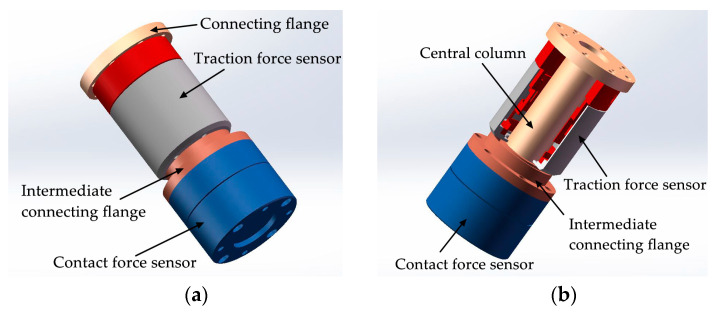
Schematic diagram of the developed tandem force sensor: (**a**) structure of the developed tandem force sensor; (**b**) the inner structure of the developed tandem force sensor.

**Figure 5 sensors-20-06042-f005:**
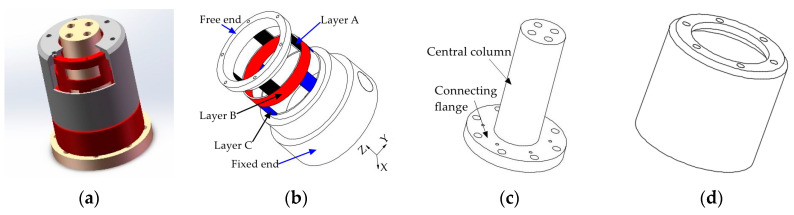
Schematic diagram of the composition of the cylindrical traction force sensor: (**a**) the basic architecture of the cylindrical traction force sensor; (**b**) cylinder-shaped elastic structural body; (**c**) connecting fitting; (**d**) shell.

**Figure 6 sensors-20-06042-f006:**
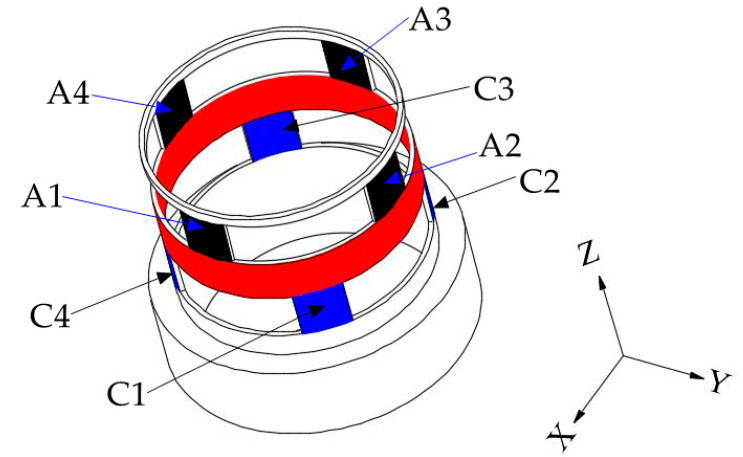
Basic structure of the cylinder-shaped elastic structural body.

**Figure 7 sensors-20-06042-f007:**
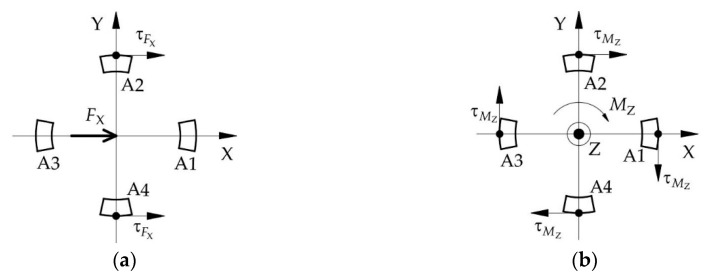
The shear stress’ direction of the points in the outside surface of layer A: (**a**) under the force *F_X_*; (**b**) under the torque *M_Z_*.

**Figure 8 sensors-20-06042-f008:**
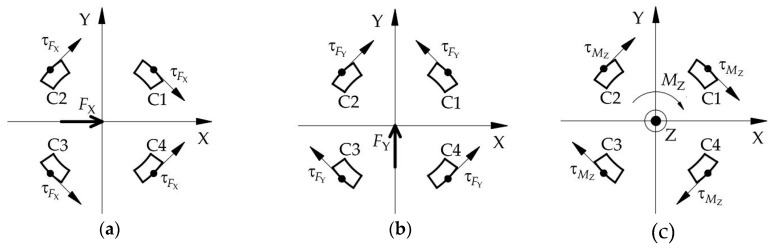
The shear stress’ direction of the points in the outside surface of layer C: (**a**) under the force *F_X_*; (**b**) under the force *F_Y_*; (**c**) under the torque *M_Z_*.

**Figure 9 sensors-20-06042-f009:**
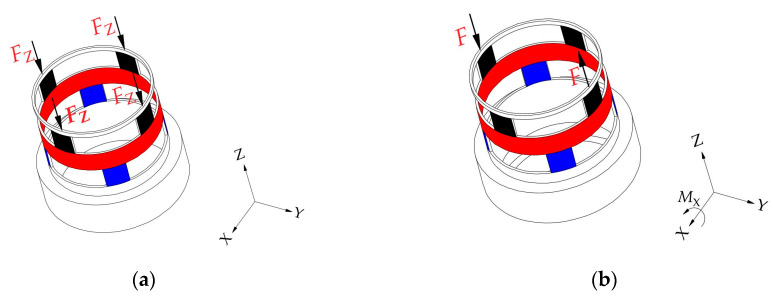
Force diagram of the cylindrical traction force sensor: (**a**) under the force *F_Z_*; (**b**) under the torque *M_X_*.

**Figure 10 sensors-20-06042-f010:**
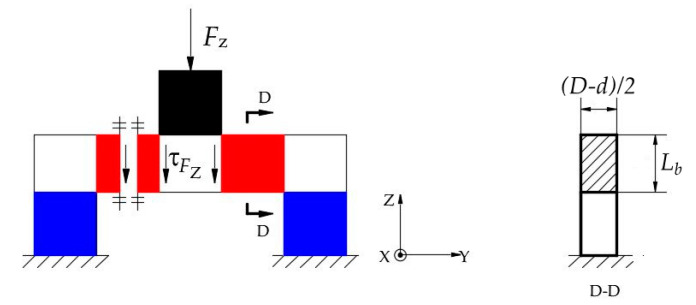
Basic constitutional unit of cylinder-shaped elastic structural body.

**Figure 11 sensors-20-06042-f011:**
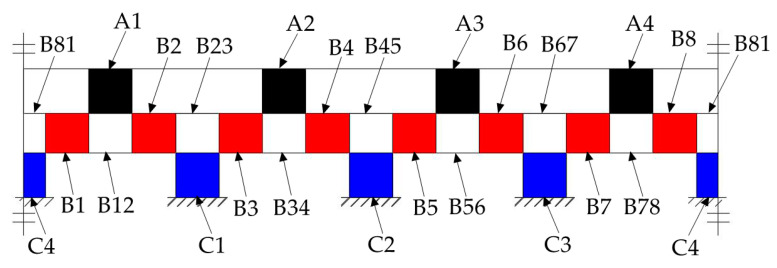
The unfold of the cylinder-shaped elastic structural body.

**Figure 12 sensors-20-06042-f012:**
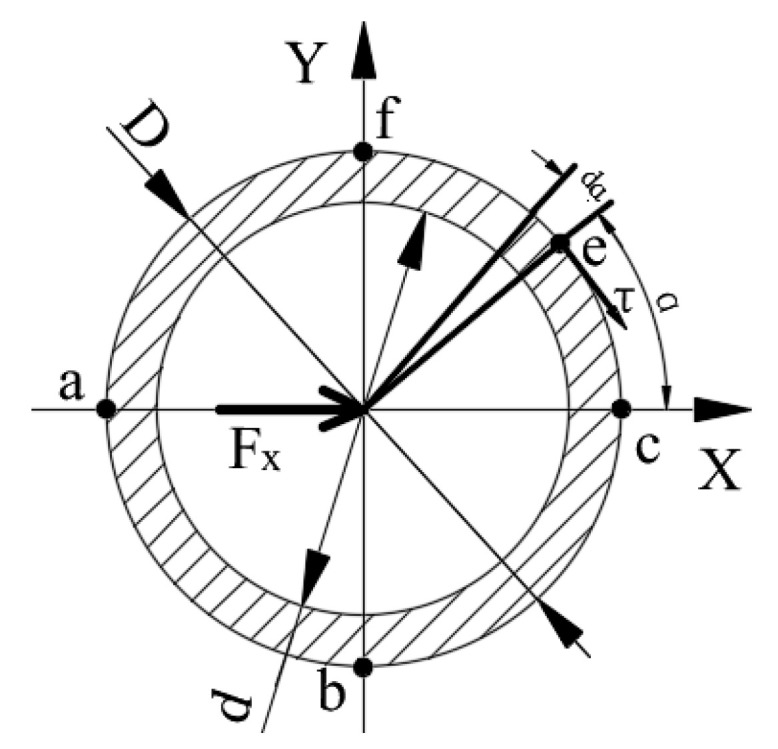
The shear stress analysis of the circular ring.

**Figure 13 sensors-20-06042-f013:**
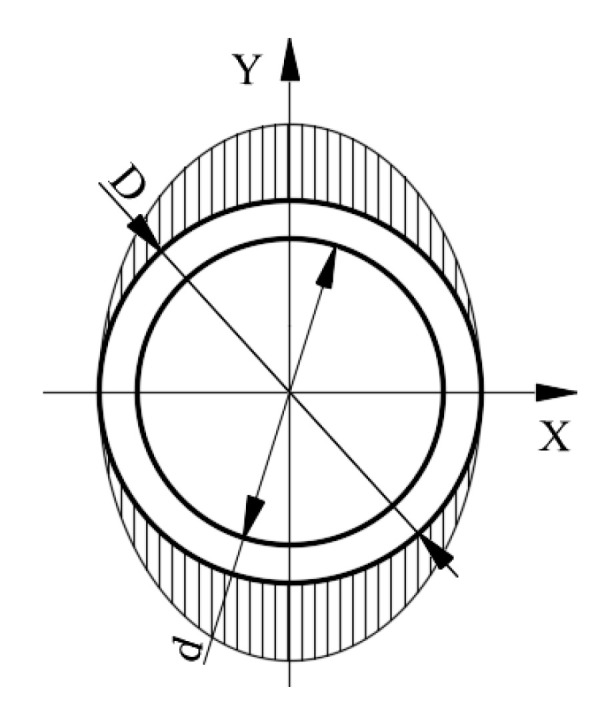
The distribution of shear stress values of the points in the outside surface of the circular ring.

**Figure 14 sensors-20-06042-f014:**
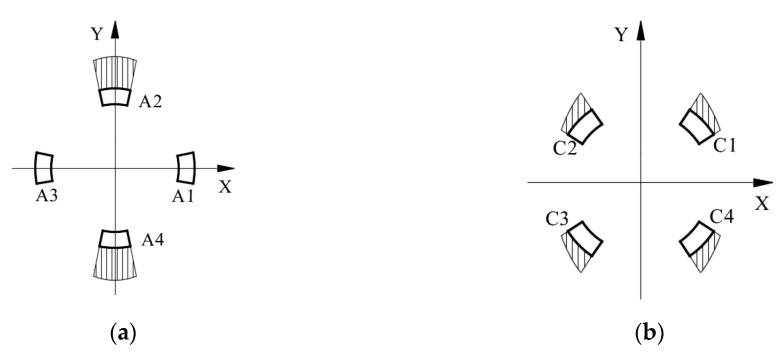
The distribution of shear stress values of the points in the outside surface of layer A and layer C: (**a**) layer A; (**b**) layer C.

**Figure 15 sensors-20-06042-f015:**
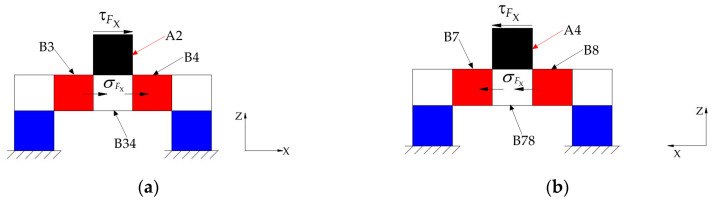
The stress in layer B induced by *F_X_*: (**a**) the stress transmitted to B34 by A2; (**b**) the stress transmitted to B78 by A4.

**Figure 16 sensors-20-06042-f016:**
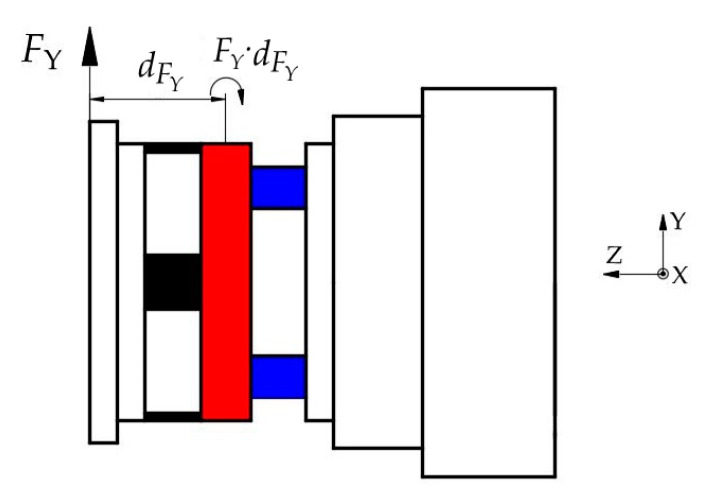
The moment applied on layer B caused by *F_Y_*.

**Figure 17 sensors-20-06042-f017:**
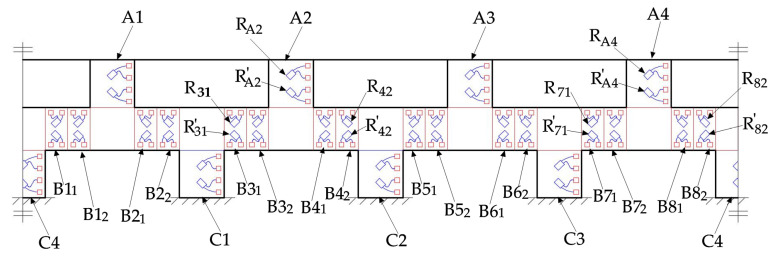
Distribution of strain gauges pasted on the cylinder-shaped elastic structural body.

**Figure 18 sensors-20-06042-f018:**
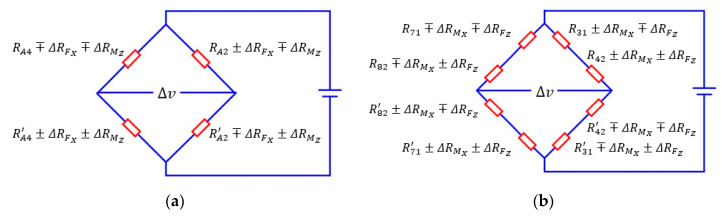
The connection mode of strain gauges in electric bridge: (**a**) the first electric bridge; (**b**) the fourth electric bridge.

**Figure 19 sensors-20-06042-f019:**
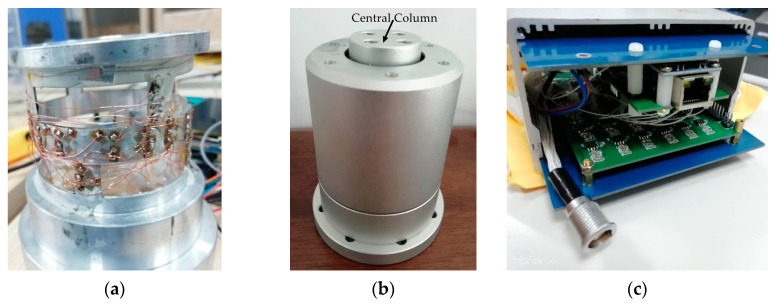
Cylinder-shaped elastic structural body and traction force sensor: (**a**) cylinder-shaped elastic structural body; (**b**) traction force sensor; (**c**) signal acquisition instrument.

**Figure 20 sensors-20-06042-f020:**
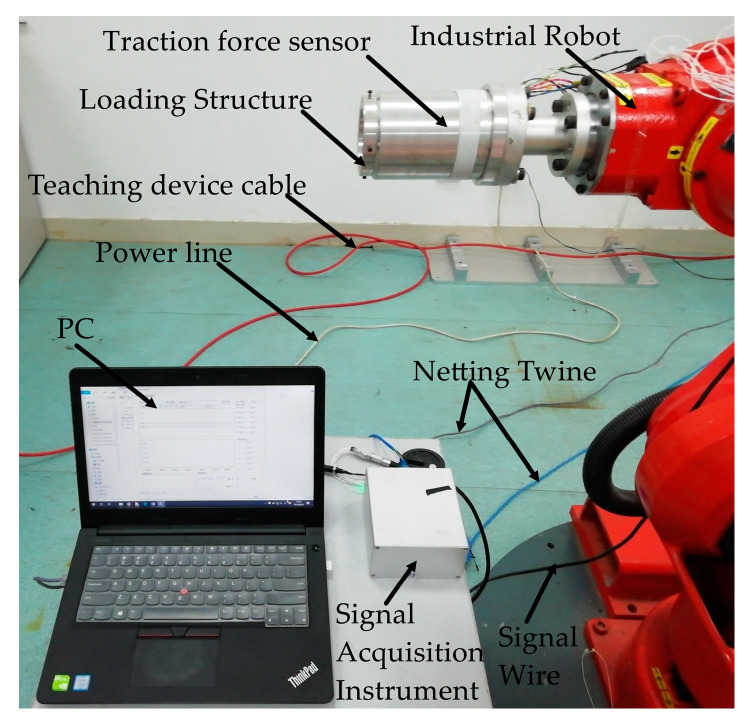
Industrial robot used in calibration experiments.

**Figure 21 sensors-20-06042-f021:**
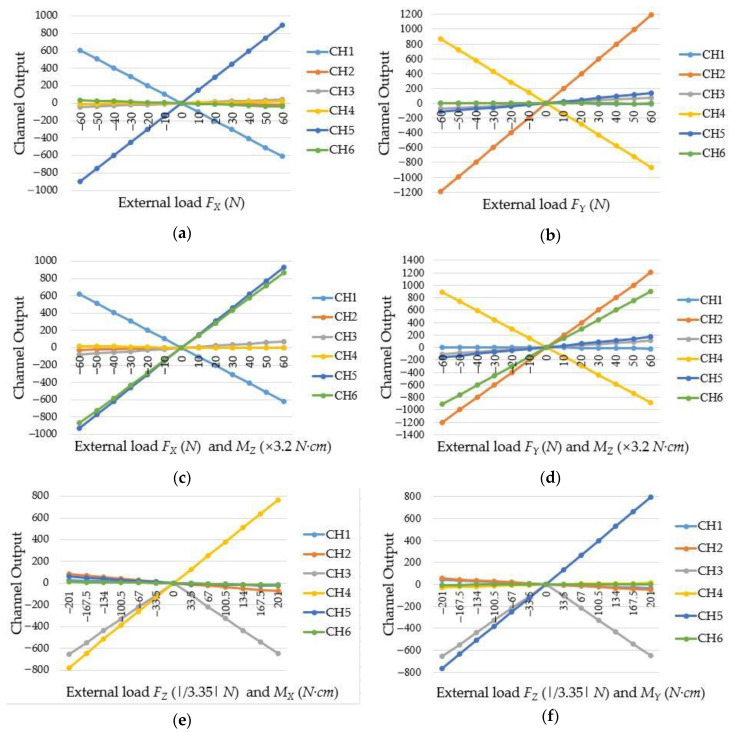
The output voltage changes of electric bridges: (**a**) under the force *F_X_*; (**b**) under the force *F_Y_*; (**c**) under the force *F_X_* and moment *M_Z_*; (**d**) under the force *F_Y_* and moment *M_Z_*; (**e**) under the force *F_Z_* and moment *M_X_*; (**f**) under the force *F_Z_* and moment *M_Y_*.

**Figure 22 sensors-20-06042-f022:**
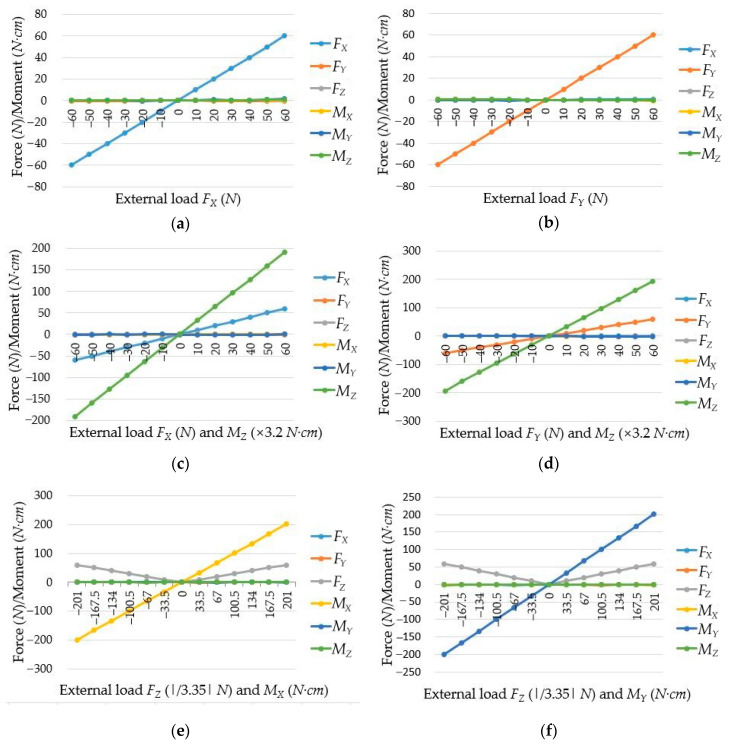
Force/Torque obtained by the cylindrical traction force sensor: (**a**) the force *F_X_* acts on the sensor; (**b**) the force *F_Y_* acts on the sensor; (**c**) under the force *F_X_* and torque *M_Z_*; (**d**) under the force *F_Y_* and torque *M_Z_*; (**e**) under the force *F_Z_* and torque *M_X_*; (**f**) under the force *F_Z_* and torque *M_Y_*.

**Figure 23 sensors-20-06042-f023:**
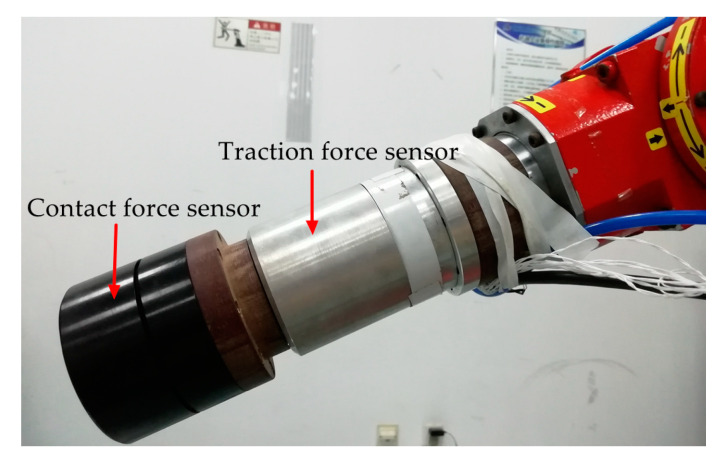
The developed tandem force sensor.

**Figure 24 sensors-20-06042-f024:**
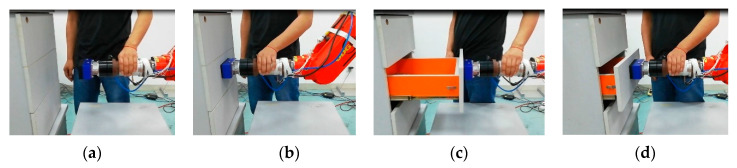
Human-robot cooperate to finish drawer switch experiment: (**a**) approach to the target; (**b**) grab the target; (**c**) open switch (**d**) close switch.

**Figure 25 sensors-20-06042-f025:**
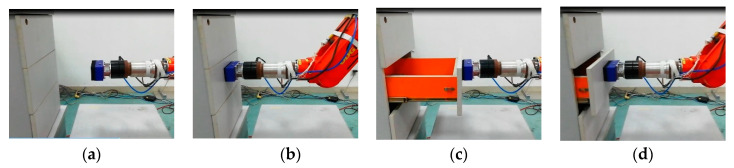
Robot finish drawer switch with the method human teaches: (**a**) approach to the target; (**b**) grab the target; (**c**) open switch; (**d**) close switch.

**Figure 26 sensors-20-06042-f026:**
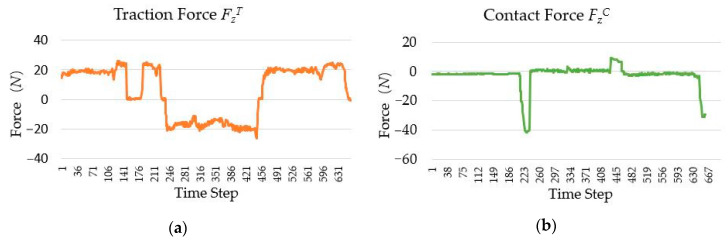
The change curve of the traction and contact forces in the drawer switch experiment: (**a**) Traction force $F_{Z}^{T}$; (**b**) contact force $F_{Z}^{C}$.

**Figure 27 sensors-20-06042-f027:**
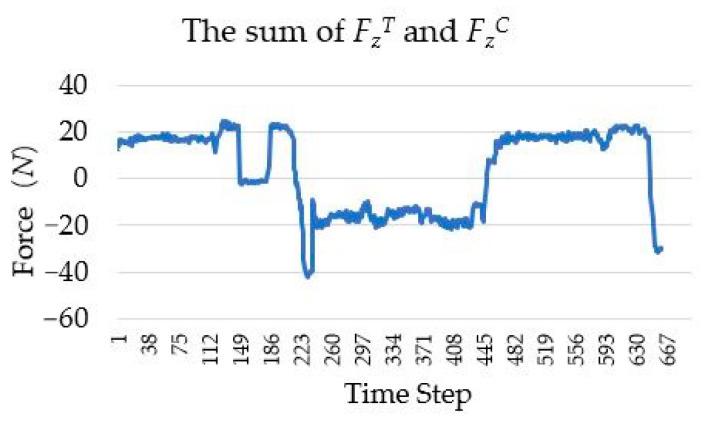
The change curve of the sum of the traction force $F_{Z}^{T}$ and the contact force $F_{Z}^{C}$.

**Table 1 sensors-20-06042-t001:** Theoretical sensitivity of the elastic structural body.

Forces/Moments	Sensitivity	Unit
*F_X_*	0.177	με/N
*F_Y_*	0.177	με/N
*F_Z_*	1.549	με/N
*M_X_*	0.155	με/N·cm
*M_Y_*	0.155	με/N·cm
*M_Z_*	0.142	με/N·cm

**Table 2 sensors-20-06042-t002:** Interference error of the cylindrical traction force sensor.

Interference Error (%F.S.)	*F_X_*	*F_Y_*	*F_Z_*	*M_X_*	*M_Y_*	***M_Z_***
*F_X_*	-	1.13	0.14	0.03	2.17	1.20
*F_Y_*	1.09	-	0.22	0.94	0.14	1.52
*F_Z_*	0.08	0.11	-	1.10	0.86	0.10
*M_X_*	0.05	0.05	-	-	2.94×10^−4	1.54×10^−5
*M_Y_*	0.02	0.03	-	0.05	-	8.53×10^−5
*M_Z_*	-	0.35	0.14	3.43×10^−4	2.58×10^−4	-

**Table 3 sensors-20-06042-t003:** Static performance indices of the cylindrical traction force sensor.

Forces/Moments	*F_X_*	*F_Y_*	*F_Z_*	*M_X_*	*M_Y_*	*M_Z_*
**Non-Linear Error (%F.S.)**	0.02	0.11	1.54	0.46	0.29	0.69
**Hysteresis Error (%F.S.)**	0.19	0.24	2.19	0.37	0.83	0.34
**Repeatability Error (%F.S.)**	0.30	0.51	0.75	1.24	1.98	0.31

**Table 4 sensors-20-06042-t004:** Calculated and real values when forces/torques are applied on traction force sensor.

	*F_X_*	*F_Y_*	*F_Z_*	*M_X_*	*M_Y_*	*M_Z_*
Applied	60.00	0.00	0.00	0.00	0.00	192.00
Measured	60.06	−0.56	−0.11	−0.86	0.26	191.50
Error (%F.S.)	0.10	0.93	0.18	0.43	0.13	0.26
Applied	0.00	60.00	0.00	0.00	0.00	192.00
Measured	0.55	60.07	−0.07	−0.36	0.00	193.76
Error (%F.S.)	0.92	0.12	0.12	0.22	0.17	0.92
Applied	0.00	0.00	60.00	201.00	0.00	0.00
Measured	−0.10	−0.10	59.79	201.52	1.19	−0.37
Error (%F.S.)	0.17	0.17	0.35	0.25	0.57	0.19
Applied	0.00	0.00	60.00	0.00	201.00	0.00
Measured	−0.03	−0.03	59.40	0.26	201.03	−0.10
Error (%F.S.)	0.05	0.05	1.00	0.12	0.01	0.05

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
