# Peer review of "Development and Application of a Tandem Force Sensor"

_sensors, 2020, doi:10.3390/s20216042_

Round 1

Reviewer 1 Report

In this manuscript, a cylindrical traction force sensor was developed to detect the force and moment, which was connected with the wrist force sensor to form a tandem force sensor. The structure design is interesting and demonstrated to be feasible. There are some comments need to be responded:

  1. In section 2, an ideal tandem force sensor was introduced. Is it necessary? Is “absence” in the first sentence right?
  2. The models in Fig 2 (b)-(d) should be coloured like that in Fig 2 (a). Then it’s easy for readers to read.
  3. Is it necessary to place Fig 3(b) that is similar to Fig. 4. Actually, there is no any specific introduction to this subfigure.
  4. In section 3.2, the force measurement principle is explained. Many symbolic variables are introduced. In order to avoid confusion, it’s suggested to add more intuitive pictures to present the principle.
  5. Line 142, “My” is wrong.
  6. Line 185, Layer B should be layer C
  7. In section 3.3.3. and 3.3.4, it’s suggested to add a diagram of the shear stress distribution as Fig7.
  8. Fig 11 and 12 are not clear.
  9. In table 4, when no force or no moment was applied, the measured value was not zero. How to calculate the Error (F.S%)?

Author Response

Thank you for your comments concerning our manuscript entitled “Development and application of a tandem force sensor” (ID: sensors -927609). Those comments are all valuable and very helpful for revising and improving our paper. We have studied comments carefully and have made correction which we hope meet with approval. Revised portion are marked in red in the paper. The responds to the comments are in the attachment. Please see the attachment.

Reviewer 2 Report

The paper describes the development of a cylindrical force/torque sensor with 6DOF, which is designed to be coupled with a standard planar F/T sensor in order to achieve what the authors called a "tandem" sensor. The application domain for such a tandem sensor is industrial robot human hand guidance/teaching.

The proposed sensor would actually be very useful for such applications, so I believe that the content of the paper would be very interesting for industrial roboticists and practitioners.

However, an accurate revision of the English language is required to improve the quality of the paper. Examples:

  • the use of "And" at the beginning of a sentence must be avoided (e.g. line 62)
  • The beginning of line 72 is meaningless (please rewrite the whole sentence)
  • the last sentence of Section 2 is not well written. Since it seems crucial in defining the main motivation behind the choice of a cylindrical sensor, please rewrite it.
  • the term "Under" should be written in lowercase at lines 116 and 129
  • switch the words "mechanics" and "theoretical" at line 149-150
  • line 163: "show" -> "shown"
  • line 167: "angel" -> "angle"

Author Response

(The authors gave the same response as above.)

Reviewer 3 Report

According to Authors declarations, the paper presents a “tandem force sensor”; as a matter of fact, the paper is mostly devoted to the description and design of a “traction force sensor” (by Authors definition) and then, in the experimental part (last section), an unspecified force contact sensor is added to obtain a tandem (?) sensor.

Many key aspects of the paper are really unclear and make it difficult to provide a judgement of the real paper value, so, although due to the present form of the paper, it would merit a “reject”, I suggest a major revision to provide the Authors with a chance to improve their work.

In the following, a list of inconsistencies, unclear aspects or undemonstrated results:

  • a key aspect of the discussion is unclear: apparently, the traction sensor is provided with a (mostly rigid) central column (fig. 2-c, fig. 13-b) to which the next component is attached (e.g., the contact force sensor): the Authors do not clarify how external forces are applied to the deformable part of their traction force sensor and then measured. A clear explanation on how the sensor is built and its parts connected between them and with external components is mandatory. A clear and detailed scheme of the tandem force sensor in which it is graphically clarified how forces and torques “flow” through the sensor would be very helpful
  • as a consequence of previous observation, the Authors should be clearly explain how their proposed traction force sensor is different (and possibly better) than a normal  6-dof force sensor
  • Fig. 1-a (a continuous lateral sensitive surface) is inconsistent with Fig. 2-b in which the cylinder shaped elastomer is discontinuous
  • Section 3.3 contains a lot of relations used to design the traction sensor, that are introduced without any justification or proof, so the correctness of the provided results is unproved
  • in Section 3.4.2 the Authors reason about the “maximum force that the cylinder-shaped elastomer” can withstand, but then they cite “the permissible shear stress of 7075 aluminium alloy” (not an elastomer ...)
  • Section 6.6 which should be one of the key points of the paper is almost incomprehensible and the scheme in Fig. 8 should be much better documented and represented
  • one would expect that a “traction force sensor” would measure only a (positive) axial force, but apparently the discussed sensor measures 6 forces/torques both positive and negative, so the proposed name is not suited
  • English language should be significantly improved (see for example the first phrase in Section 3.3)

Author Response

Thank you for your comments concerning our manuscript entitled “Development and application of a tandem force sensor” (ID: Sensors -927609). Those comments are all valuable and very helpful for revising and improving our paper. We have studied comments carefully and have made correction which we hope meet with approval. Revised portion are marked in red in the paper. The responds to the comments are in the attachment. Please see the attachment.

Round 2

Reviewer 3 Report

The paper presents a perceptual system based on two sensors, one for traction and one for contact force, to be used in robotic applications.

Although the paper may have some merits, there are key aspects that should be improved before the paper can be published:

The Authors affirm that the two sensors are "in series", but as clearly shown in the scheme in Fig. 2 and in Fig. 3-b, the two sensors act in parallel: contact force acts on the end-effector and both is detected dependently of traction force and does not flow through the sensible part of the traction force sensor. Traction force does not flow through the contact force "sensible" part.

As a matter of fact, the paper presents the detailed characteristics of a “traction force” sensor, and then in the experimental part, they use it (in PARALLEL) with a generic end-effector force sensor (detecting contact force).

My suggestion is that the paper is rewritten clearly highlighting  the true nature of the presented couple of sensors and of their integration (in fact, minimal integration), and showing how their use introduce innovation with respect to any couple of (in parallel) force sensors.

Finally, the Authors use extensively the word "elastomer", then the affirm that:" ...  Aluminum alloy 7075 is selected to machine the cylinder-shaped elastomer ...", but  Aluminum alloy 7075 is not an elastomer (see below). So I cannot understand why they continuously use the word “elastomer”.

A couple of definitions of "elastomer":

"Elastomers (rubbers) are special polymers that are very elastic"

(https://polymerdatabase.com/Elastomers/Elastomers.html)

An Elastomer is a polymer with viscoelasticity (i.e., both viscosity and elasticity) and has very weak intermolecular forces

https://en.wikipedia.org/wiki/Elastomer 

Author Response

Thank you for your constructive comments concerning manuscript entitled “Development and application of a tandem force sensor” (ID: Sensors -927609). Those comments are of great significance for improving this paper. We have studied comments carefully and have made correction which we hope meet with approval. Revised portion are marked in red in the paper.

The main corrections in the paper and the responds to the comments are in the attachment. Please see the attachment.
